# EXPLAINING TO LEARN:
# REGULARIZATION USING CONTRASTIVE VISUAL EXPLANATION PAIRS FOR DISTRIBUTION SHIFTS

## ABSTRACT

While a myriad of algorithms have been proposed to address distribution shifts, most algorithms are known to perform best only under specific conditions and fail to outperform the baseline empirical risk minimization (ERM) in other scenarios. Furthermore, the algorithmic complexity of some existing methods can render them less interpretable, and their approach to addressing spurious correlations—a hallmark of distribution shifts—is often indirect. To specifically address spatial confounders, we propose Explaining to Learn (ETL), an interpretable, explanation-based learning algorithm that removes spatial confounders from the primary classifier's latent representations during training. ETL achieves this by penalizing the similarity between GradCAM activation maps from a primary label classifier and a concurrently trained confounder classifier. On the more recent and difficult *Spawrious Many-to-Many Hard Challenge* benchmark, ETL achieves an average accuracy (AA) of 82.24% (±3.87) and a worst-group accuracy (WGA) of 66.31% (±8.73), outperforming the leading state-of-the-art (SOTA) benchmark by a significant 5% and 11%, respectively. This strong performance extends to other challenging benchmarks, where ETL also outperforms SOTA regularization methods on *CMNIST* (AA: 69.02% ±0.53; WGA: 67.63% ±1.39) and *Waterbirds* (AA: 92.12% ±0.67; WGA: 86.92% ±0.56). We complement these empirical results with theoretical analyses, demonstrating the viability of explanation-based learning for mitigating distribution shifts.

## 1 INTRODUCTION

Standard machine learning approaches depend on the assumption that training and test data are independently and identically distributed (i.i.d.) (Ye et al., 2021), and if this assumption is violated, this leads to issues where the model may underperform on the test data or depend on spurious correlations in the training data (Monga et al., 2025; Suhail & Sethi, 2025; Koh et al., 2020), which are not conceptually predictive of the labels.

This assumption is normally violated in the contexts of distribution shifts which are divided into two types: subpopulation shifts and domain generalization (Koh et al., 2020). Subpopulation shifts happen when the proportions of subpopulations, such as demographic or other contextual variables, change from training to test (Koh et al., 2020; Yang et al., 2023). To understand domain generalization, domains, otherwise known as environments, are generally defined as data distributions with unique statistical characteristics and conditions (Koh et al., 2020; Ye et al., 2021). Domain generalization is then defined as learning only from source domain/s during training and learning to generalize to unseen target domains at deployment (Koh et al., 2020; Ye et al., 2021).

While several regularization algorithms have been developed for both, the problem is that most methods do not specifically address spurious correlations within the training data, which is mostly the cause of the model underperformance (Wiles et al., 2022). Moreover, while theoretically elegant, the mechanisms in most algorithms, such as DANN's gradient reversal (Ganin et al., 2016), are not intuitively explainable to laypersons, leading to potential issues with gaining stakeholder trust and support.

Given these current gaps, the study proposes a novel intersectional algorithm in the fields of Distribution Shifts and Explainable AI (xAI). The study introduces Explaining to Learn (ETL), an algorithm which penalizes the similarity in the gradient-weighted class activation maps (GradCAM) between a model being trained on the class labels and a model being trained on *a priori* confounders during training. Through this method of using an explainability technique during training which addresses spurious correlations and promotes domain invariance, the algorithm seeks to garner better distribution shift performance gains in comparison to existing SOTA algorithms and be also highly interpretable.

## 2 RELATED WORKS

**Distribution Shifts.** Minimax fairness and direct risk minimization for the worst-case group have often been the main strategy towards handling subpopulation shifts(Koh et al., 2020; Sagawa* et al., 2020; Yang et al., 2023). Still, this strategy often comes at a cost of lower overall model performance (Shen & Zhao, 2025). On the other hand, several theoretical analyses have been done on domain generalization where domain invariance of feature representations is designated as the primary solution (Liu, et. al., 2023). Despite these theoretical advances, these algorithms do not still empirically outperform the ERM baseline in general, especially on real-world image data (Wiles et al., 2022; Gulrajani & Lopez-Paz, 2021). Accounting for this, Wiles et al. (2022) proposed a newer perspective to the problem, where the authors decomposed the label attribute space into label and nuisance attributes. which simplifies the problem of having differences across domains.

**Spurious Correlations.** In terms of addressing spurious correlations, Sagawa* et al. (2020)'s GroupDRO still remains the state-of-the-art regularization algorithm, outperforming ERM with *Uniform Group Sampling* in terms of worst-group accuracy, although its removal of spurious correlations is indirect as it relies on prioritization of worst-performing groups, to achieve it.

**Explanation-based Learning.** Adebayo et al. (2018) reviewed current explainability approaches and posited that methods such as backpropagation methods produce the same explanations regardless of network reparametrizations. They differentiated these with gradient-based methods such as GradCAM, where they posited that GradCAM provides more faithful, saliency maps which can be used to debug its network (Adebayo et al., 2018). To the best of their abilities, the authors have not found studies which directly compares two GradCAM maps from different classifiers trained on the same data but different labels at training time. Most studies either use GradCAM as a part of the data curation process or only compare GradCAM maps to existing synthetic ground truth patch masks (Dammu & Shah, 2023; Hagos et al., 2022).

## 3 THEORETICAL FRAMEWORK

### 3.1 PROBLEM SETTING

The proposed algorithm is introduced by the following setup: there is a set of input image tensors, $\mathbb{X} = \{\mathbf{X}_1, \mathbf{X}_2, \ldots, \mathbf{X}_n\}$, a set of labels, $\mathbb{Y} = \{y_1, \ldots, y_{n_c}\}$, where $n_y$ is the number of unique classes, and a set of *a priori* environmental confounding variables, $E = \{e_1, \ldots, e_{n_e}\}$ where $n_e$ is the number of unique confounders. While existing algorithms have used the concept of environments in their setup, the study's method will be making use instead of Sagawa* et al. (2020)'s group setup to take advantage of the group setting in addressing spurious correlations. Given that $\mathbb{X} \to \mathbb{Y}$ and $\mathbb{X} \to E$, each input tensor can be categorized into groups using their corresponding confounder and label variables, such that, $\mathbb{G} = \{g_1, \ldots, g_{|\gamma|} : \gamma \in E \times \mathbb{Y}\}$ where $\exists g = \varnothing$ and $g = \{\mathbf{X}_1, \ldots, \mathbf{X}_{|g|}\}$.

For the primary label classification task, we then now construct the latent representation function for the label, $h_y : \mathbf{X} \to Z$ where $Z = \{z_1, z_2, \ldots, z_n\}$ and the label classifier function, $f_y : Z \to \mathbb{Y}$. In the distribution shift and spurious correlation setting, given that $P(\mathbb{G}_{train}) \neq P(\mathbb{G}_{test})$ where $P(\mathbb{G}) = P(\mathbb{Y}, E)$ and $P(Z \mid \mathbf{X}_{train}, \mathbb{Y}_{train}, E_{train}) \neq P(Z \mid \mathbf{X}_{test}, Y_{test})$ which leads consequently to $P(\hat{\mathbb{Y}} \mid Z, \mathbb{X}_{train}, \mathbb{Y}_{train}, E_{train}) \neq P(\hat{\mathbb{Y}} \mid Z, \mathbb{X}_{test}, \mathbb{Y}_{test})$, we can further denote that in the presence of confounders $E$ within each group, each latent representation can be decomposed into invariant features and environmental confounding features, such that, $Z = Z_{inv} + Z_{env}$, where invariant features are conceptually predictive of the labels while confounding features are features which are more conceptually predictive of the groups and only coincidentally correlated to the labels

(Ming et al., 2022). The presence of this $Z_{env}$ can lead to performance decays at deployment, as these may be only be predictive during training but not during deployment (Ming et al., 2022).

Given this, the study proposes the removal of these confounding features by using the confounder's latent representation function, $h_e : \mathbb{X} \to Z_e$ and using its classifier function, $f_e : Z_e \to E$, in conjunction with the initially constructed functions, $h_c$ and $f_c$. Assuming $Z_e \approx Z_{env}$, while simply backpropagating a similarity function, $S$, on both $Z$ and $Z_e$, may intuitively solve the problem of removing the confounders $Z_{env}$ in $Z$, the problem with this direct approach is that this approach is not target-specific and may not specifically target the areas used in predicting the actual label or actual confounder.

Borrowing from Selvaraju et al. (2019)'s definitions, to make the penalty target-specific, the gradient of the logit output $\hat{t}$ with respect to each activation map $A^k$, where $\hat{t}$ corresponds to the logit output for the true target $t$ and where $t \in T$ and $T = \{t : t \in \mathbb{Y} \cup E\}$, may be used to produce more target-specific neuron importance weights $\alpha_k^{\hat{t}}$ for each $A^k$, such that the sum of all gradients per $A^k$ are averaged to get $\alpha_k^{\hat{t}}$ (Selvaraju et al., 2019):

$$\alpha_k^{\hat{t}} = \frac{1}{P} \sum_i \sum_j \frac{\partial \hat{t}}{\partial A_{i,j}^k} \tag{1}$$

where $A_{i,j}^k$ is a pixel in $A^k$ and $P$ is the total number of pixels in each activation map.

We can now make the activation maps $A^k$ more target-discriminative by multiplying the activation maps to their corresponding neuron importance weights $\alpha_k^{\hat{t}}$ (Selvaraju et al., 2019). Completing the whole gradient-weighted activation map (GradCAM) definition, Selvaraju et al. (2019) recommends performing a $ReLU$ operation to the linear combination of these products, to remove nontarget-related features in the resulting heatmap $L_{GradCAM}^{\hat{t}}$, such that:

$$L_{GradCAM}^{\hat{t}} = ReLU(\sum_k \alpha_k^{\hat{t}} A^k) \tag{2}$$

Hence, through GradCAM, we are able to compute attributions which are target-specific and whose similarity penalties would aid with achieving the primary objective of removing the confounders from the latent representation of the label classification task.

## 3.2 Proposed Algorithm

Given that we can now remove similar areas of both the actual label and confounder through Grad-CAM similarity backpropagation, we now define the general similarity loss function of ETL as:

$$l_{sim} = S(L_{GradCAM}^{\hat{y}}, L_{GradCAM}^{\hat{e}}) \tag{3}$$

where the gradients used are with respect only to each activation map $A^k$, not also to the logit's gradients, since using first-order gradients is more computationally economical to compute (Shi et al., 2021):

$$\nabla_{A_{i,j}^k} l_{sim} = \frac{\partial S(L_{GradCAM}^{\hat{y}}, L_{GradCAM}^{\hat{e}})}{\partial A_{i,j}^k} \tag{4}$$

This $l_{sim}$, along with the confounder classifier's general loss function, is then added to the label classifier's general loss function with the regularization factors $\lambda_{sim}$ and $\lambda_e$, forming ETL's training objective function $\hat{R}_{ETL}$:

$$\hat{R}_{ETL}(\theta_c, \theta_e) = \mathbb{E}_{(\mathbf{X}, y, e) \sim D_{train}}[l(\mathbf{X}, y; \theta_c) + \lambda_e l(\mathbf{X}, e; \theta_e) + \lambda_{sim} l_{sim}(\mathbf{X}, y, e; \theta_c, \theta_e)] \tag{5}$$

where $D_{train}$ is the training distribution and $\theta_c$ and $\theta_e$ are the parameters of the label model $m_y : f_y \circ h_y$ and confounder model $m_e : f_e \circ h_e$, respectively, with the primary goal of finding the optimal

$\theta_c^*$ which maximizes the label model's worst-group-accuracy (WGA), a gold standard metric for spurious correlation settings (Yang et al., 2023), within the hypothetical parameter space $\Theta$, at test time:

$$\theta_c^* = \arg\max_{\theta_c \in \Theta} \min_g \mathbb{E}_{(\mathbf{X},y) \sim D_{test}^g}[\mathbf{1}_{\hat{y}_{\theta_c} = y}] \tag{6}$$

Lastly, two kinds of sampling are employed for ETL, namely, *Random Sampling*, which randomly samples the dataset without replacement, based on the prevailing training distribution and *Uniform Group Sampling*, which tries to sample $(\mathbf{X}, y, e)$ uniformly from each $g : g \neq \varnothing$ (Sagawa* et al., 2020; Shen & Zhao, 2025).[1] It is inferred that the *Uniform Group Sampling* would be more advantageous for ETL, as it would provide a non-biased view of the data, removing inherent label-confounder proportions within the training distribution as a factor for learning spurious correlations.

## 3.3 THEORETICAL ANALYSIS

To prove the viability of ETL's $l_{sim}$, we now demonstrate its learning stability. Our reasoning proceeds from a set of mild assumptions to show that the loss function is *Lipschitz continuous* with respect to the model parameters, which is a key condition for stable gradient-based optimization. (See A.2 for the complete proof).

**Assumption 1** *There exists an nth-indexed final-layer weight parameter $w_n^{\hat{t}_i}$ corresponding to a target logit $\hat{t}_i$ which is not equal to the nth-indexed final-layer weight parameter $w_n^{\hat{t}_j}$ corresponding to another target logit $\hat{t}_j$.*

$$\exists w_n^{\hat{t}_i} : w_n^{\hat{t}_i} \neq w_n^{\hat{t}_j}; i, j \in \{1, \ldots, |T|\}; i \neq j \tag{7}$$

Given Assumption 1 and a single input tensor $\mathbf{X}$, by decomposing $\alpha_k^{\hat{t}}$ into $\frac{1}{P}$ and $w_k^{\hat{t}}$ using the Chain Rule, we can prove that the GradCAM maps of two logits $\hat{t}_i$ and $\hat{t}_j$ from the same classifier are also unequal.

**Lemma 1 (GradCAM Target-Specificity)** *Given a single input tensor $\mathbf{X}$ and a model $m$ and using the chain rule on $\alpha_k^{\hat{t}_n}$, the GradCAM maps of any two logit from the same model and input are unequal.*

$$ReLU(\frac{1}{P^2} \sum_k w_k^{\hat{t}_i} A^k) \neq ReLU(\frac{1}{P^2} \sum_k w_k^{\hat{t}_j} A^k) \tag{8}$$

Placing this in the label-confounder classifier pair setting, we assume the following:

**Assumption 2** *For a given input $\mathbf{X}$, the activation maps produced by the label model's feature extractor, $A_y^k$, and the confounder model's feature extractor, $A_e^k$, are not identical.*

$$A_y^k(\mathbf{X}) \neq A_e^k(\mathbf{X}) \tag{9}$$

Using Assumption 3, we can then extend Lemma 1 to the label-confounder classifier pair setting:

**Corollary 1 (GradCAM Target-Specificity Extension)** *Extending Lemma 1 for two models with Assumption 2, the GradCAM maps of any two logits each from the label $y$ and confounder $e$ model are unequal.*

$$ReLU(\frac{1}{P^2} \sum_k w_k^{\hat{y}} A_y^k) \neq ReLU(\frac{1}{P^2} \sum_k w_k^{\hat{e}} A_e^k) \tag{10}$$

---

[1] Another sampling method used in distribution shifts is *Uniform Environment Sampling*, which will not be used due to $\hat{R}_{ETL}$ not being directly applicable to environment settings.

Given that we have established that GradCAM maps of any label logit $\hat{y}$ and confounder logit $e$ are unequal and are proportional to the logit final-layer weights, hence highlighting activation map importance, we can now define their similarity function.

**Definition 1 (GradCAM Similarity Function)** *A GradCAM Similarity Function, $S$, is a function which takes two inputs, GradCAM maps ($M_y$, $M_e$) from a label and classifier model. This function is L-Lipschitz continuous and differentiable for all $x$, or at least differentiable for a given interval and subdifferentiable at a given $x$.*

$$GradCAM\ Similarity = S(M_y, M_e) \tag{11}$$

Using this definition, we can then check the function's stability using the $L\text{-}Lipschitz\ continuity$ equation.

**Theorem 1 (Similarity Function Loss Stability)** *Given Definition 2, we define the similarity function loss as $l_{sim}(\theta) = S(M_y, M_e)$ where $\theta$ is the model parameters, and using $L\text{-}Lipschitz$ continuity equation, we can prove that:*

$$l_{sim}(\theta) - l_{sim}(\theta') \leqslant L\sqrt{C_y^2 + C_e^2} \cdot \|\theta - \theta'\|_2 \tag{12}$$

where $C_y^2$ and $C_e^2$ are the $Lipschitz$ constants for both $M_y$ and $M_e$, and the change in model parameters, $\|\theta - \theta'\|_2$ also causes a change in the similarity loss function, which is the definition of $Lipschitz\ continuity$ for the loss function. This property is crucial as it bounds the gradient, preventing erratic weight updates and ensuring a stable learning dynamic.

## 4 METHODOLOGY

### 4.1 DATASETS

#### 4.1.1 SUBPOPULATION SHIFTS

In accounting for subpopulation shifts, the study used standard benchmarks such as Arjovsky et al. (2020)'s *CMNIST* dataset and Sagawa* et al. (2020)'s *Waterbirds* and *CelebA* dataset. These datasets allow the authors to test ETL on more synthetic, subject-inherent spurious correlations (e.g., $E = \{\text{red, green}\}$ in *CMNIST*), on natural, background-based spurious correlations (e.g., $E = \{\text{water, land}\}$ in *Waterbirds*), and real-life, subject-inherent spurious correlations (e.g., $E = \{\text{male, female}\}$ in *CelebA*).

#### 4.1.2 DOMAIN GENERALIZATION

In accounting for domain generalization, the study used one of the latest challenging domain generalization benchmarks which involves spurious correlations, namely, the *Spawrious* benchmark (Lynch et al., 2025). To account for the full breadth of the dataset while considering budget constraints, the study used the easiest challenge, the *Spawrious One-to-One Easy* challenge, and the most difficult challenge in the dataset, the *Spawrious Many-to-Many Hard* challenge. In these challenges, the authors are able to test whether ETL is able to generalize to one-to-one shifting of background confounders (e.g., Dirt $\rightarrow$ Beach) and many-to-many shifting of background confounders corresponding to each label (e.g., Dirt, Snow $\rightarrow$ Beach, Jungle) (Lynch et al., 2025).

### 4.2 ALGORITHMS

As a general baseline, the authors used ERM. To provide a direct comparison for group-based methods geared towards subpopulation shifts, the authors used GroupDRO as the SOTA group-based baseline (Sagawa* et al., 2020). For comparisons to environment-based methods geared towards domain generalization, the authors used SOTA domain generalization methods such as IRM (Arjovsky et al., 2020), MMD (Li et al., 2018), CORAL (Sun & Saenko, 2016), DANN (Ganin et al., 2016), and CDANN (Long et al., 2018). The authors prioritized regularization methods over other methods such as data augmentation, to provide analogous comparisons to ETL.

### 4.3 Evaluation Criteria

Following Sagawa* et al. (2020) and Yang et al. (2023), the authors used both average accuracy (AA) and worst-group accuracy (WGA) as the evaluation criteria. Both are used to highlight the WGA-AA trade-off when optimizing for WGA (Sagawa* et al., 2020; Yang et al., 2023).

### 4.4 Regularization

Given the breadth of possible similarity functions, the authors started with a roster of 11 similarity functions, which uses either *random sampling* or *uniform group sampling* (22 setups all-in-all). To limit these ETL algorithms in the next runs, only the top-3 unique similarity setups in the first run of the *CMNIST* and *Waterbirds* datasets, in terms of WGA (above or equal to the 99th percentile)[2], are used in the next runs. The final functions used are the negative mean absolute error (MAE), negative Jensen-Shannon distance (JS distance), cosine similarity, structural similarity index measure (SSIM), and soft Dice.

### 4.5 Spurious Correlation & Domain Invariance Evaluation

In showing the separation of label and confounders, the authors use GradCAM plots (Selvaraju et al., 2019) to highlight the reliance or non-reliance of specific algorithms on confounders.

Furthermore, the authors used Uniform Manifold Approximation and Projection (UMAP), a scalable, nonlinear dimensionality reduction technique (McInnes et al., 2020), to plot and evaluate latent representations in terms of label, confounder, and preset environment separation. These qualitative evaluations are paired with Maximum Mean Discrepancy (MMD), a nonparametric method for quantifying the difference between probability distributions (Gretton et al., 2012), i.e., the latent space distributions.

Refer to Appendix A.1 for a more detailed discussion of the methodology.

## 5 Results and Discussion

### 5.1 Experiment Results

Table 1: CMNIST and Waterbirds Experiment Results

| Algorithms | | Subpopulation Shift | | | |
|---|---|---|---|---|---|
| | | CMNIST | | Waterbirds | |
| Name | Sampling | AA | WGA | AA | WGA |
| *Baselines* | | | | | |
| ERM | Random | 32.96 ±2.9% | 25.11 ±2.5% | 62.86 ±15.8% | 23.68 ±21.1% |
| IRM | Uniform Env. | 65.11 ±1.1% | 64.59 ±0.9% | 89.16 ±2.4% | 84.42 ±3.0% |
| MMD | Uniform Env. | 65.41 ±0.8% | 61.98 ±1.4% | 90.81 ±1.1% | 85.64 ±0.2% |
| CORAL | Uniform Env. | 63.62 ±4.1% | 59.23 ±7.0% | 90.00 ±0.8% | 85.41 ±0.8% |
| DANN | Uniform Env. | 67.32 ±1.6% | 66.42 ±1.6% | 90.71 ±0.6% | 85.41 ±0.8% |
| CDANN | Uniform Env. | 35.25 ±24.4% | 21.00 ±18.8% | 89.57 ±1.7% | 84.71 ±1.8% |
| GroupDRO | Uniform Group | 66.14 ±2.1% | 63.79 ±3.5% | 90.50 ±0.4% | 85.95 ±1.2% |
| *Ours* | | | | | |
| **ETL-MAE** | **Uniform Group** | **69.02 ±0.5%** | **67.63 ±1.4%** | **89.59 ±0.5%** | **87.45 ±0.6%** |
| ETL-Cosine | Uniform Group | 67.90 ±0.6% | 66.15 ±1.0% | 90.85 ±1.0% | 85.72 ±0.7% |
| **ETL-Soft Dice** | **Uniform Group** | 66.75 ±1.6% | 65.14 ±2.1% | **92.12 ±0.7%** | **86.92 ±0.6%** |
| **ETL-JS Dist.** | **Uniform Group** | **69.66 ±0.9%** | **66.88 ±2.3%** | 89.18 ±1.9% | 85.10 ±2.3% |
| ETL-SSIM | Uniform Group | 67.85 ±1.0% | 65.87 ±2.1% | 89.77 ±1.5% | 85.10 ±1.2% |

---

[2]This is to ensure maximal performance.

In Table 1, ETL-MAE performs consistently better than other SOTA algorithms in terms of WGA, with less variability for both *CMNIST* and *Waterbirds*. Furthermore, other similarity functions such as soft Dice and JS distance are better at achieving higher AA, than WGA. This may be linked to how MAE is more stringent in penalizing pixel-to-pixel similarities, as compared to the other two. This highlights how different similarity functions may be used depending on which metric is prioritized. Overall, ETL performed better than GroupDRO, the subpopulation shift SOTA, for both datasets (Refer to Appendix A.4 for ETL's performant results on *CelebA*).

Table 2: Spawrious Experiment Results

| Algorithms | | Domain Generalization | | | |
|---|---|---|---|---|---|
| | | Spawrious O2O (Easy) | | Spawrious M2M (Hard) | |
| **Name** | **Sampling** | **AA** | **WGA** | **AA** | **WGA** |
| *Baselines* | | | | | |
| ERM | Random | 91.00 ±1.1% | 84.12 ±2.0% | 77.37 ±2.4% | 54.91 ±2.5% |
| IRM | Uniform Env. | 93.01 ±1.3% | 86.20 ±3.1% | 64.66 ±0.5% | 35.19 ±2.3% |
| MMD | Uniform Env. | 93.13 ±1.5% | 86.17 ±2.9% | 76.64 ±3.9% | 54.35 ±4.7% |
| CORAL | Uniform Env. | 93.14 ±1.6% | 86.43 ±3.0% | 76.74 ±3.5% | 53.74 ±4.7% |
| DANN | Uniform Env. | 94.04 ±1.0% | 87.55 ±1.2% | 76.75 ±4.8% | 55.41 ±9.3% |
| CDANN | Uniform Env. | 93.43 ±1.7% | 85.24 ±2.9% | 73.95 ±9.3% | 48.23 ±12.4% |
| **GroupDRO** | **Uniform Group** | **95.13 ±0.3%** | **90.32 ±1.0%** | 77.70 ±2.2% | 54.40 ±1.4% |
| *Ours* | | | | | |
| **ETL-MAE** | **Uniform Group** | **94.30 ±1.0%** | **88.05 ±1.1%** | **82.24 ±3.9%** | **66.31 ±8.7%** |
| ETL-Cosine | Uniform Group | 93.97 ±0.3% | 87.71 ±1.0% | 81.37 ±7.5% | 64.25 ±13.3% |
| ETL-Soft Dice | Uniform Group | 91.34 ±1.2% | 81.78 ±2.1% | 76.10 ±4.5% | 54.09 ±7.8% |
| ETL-JS Dist. | Uniform Group | 93.81 ±1.2% | 87.74 ±2.5% | 60.77 ±24.1% | 38.02 ±36.1% |
| ETL-SSIM | Uniform Group | 93.52 ±1.0% | 85.96 ±2.4% | 74.16 ±6.6% | 46.10 ±11.9% |

As shown in Table 2, while GroupDRO performed better in the *Easy Spawrious* benchmark, ETL achieved an AA higher than 80% and WGA higher than 60% in the *Hard Spawrious* benchmark, which were not achieved by the other baseline algorithms. This potentially highlights the bias-variance trade-off in which ETL may provide higher variance and may be meant for more complex spurious correlation settings, given that it performs better in the presence of two unseen backgrounds per subject at test time, a key feature of the *Hard Spawrious* benchmark, in comparison to only one unseen background per subject in the *Easy Spawrious* benchmark.

## 5.2 GradCAM Comparison Studies

In Figure 1, most of the GradCAM plots of ERM and GroupDRO show signs of susceptibility to spurious correlations, in which some of the hot areas in their heatmaps lack conceptual relationship to the $y$. Additionally, it can be inferred that the reliance on these spurious correlations led to incorrect $\hat{y}$ or predictions. Clear separation between $\hat{y}$ and $e$ can be observed in the GradCAM plots of ETL, showing that $\hat{R}_{ETL}$ was able to address spatial confounders at training time. More interestingly, ETL relied on the edges for *CMNIST* as shown by the marginal hot areas, and it showed more conceptual understanding of dachshunds as shown by its focus on the ears, rather than the body. Other algorithms were not able to have an anchor for *CMNIST* due to their reliance on the other color **Green** being related to $< 5$ and focused also on the beach and fur color for *Hard Spawrious* challenge. In the case of fur color, it is shown that convolutional neural networks can even form coincidental correlations which adds up to the primary confounder, and ETL still was able to address this, by just targeting the progenitor, primary confounder.

## 5.3 UMAP Comparison Studies

To understand Figure 2, reliance on label attributes and non-reliance on confounders are evident in plots where labels are clearly separated (high $\text{MMD}_{\mathbb{Y}}$) while confounders are interspersed with each other (low $\text{MMD}_E$). While ERM mixes both labels, GroupDRO presents a dispersed label separa-

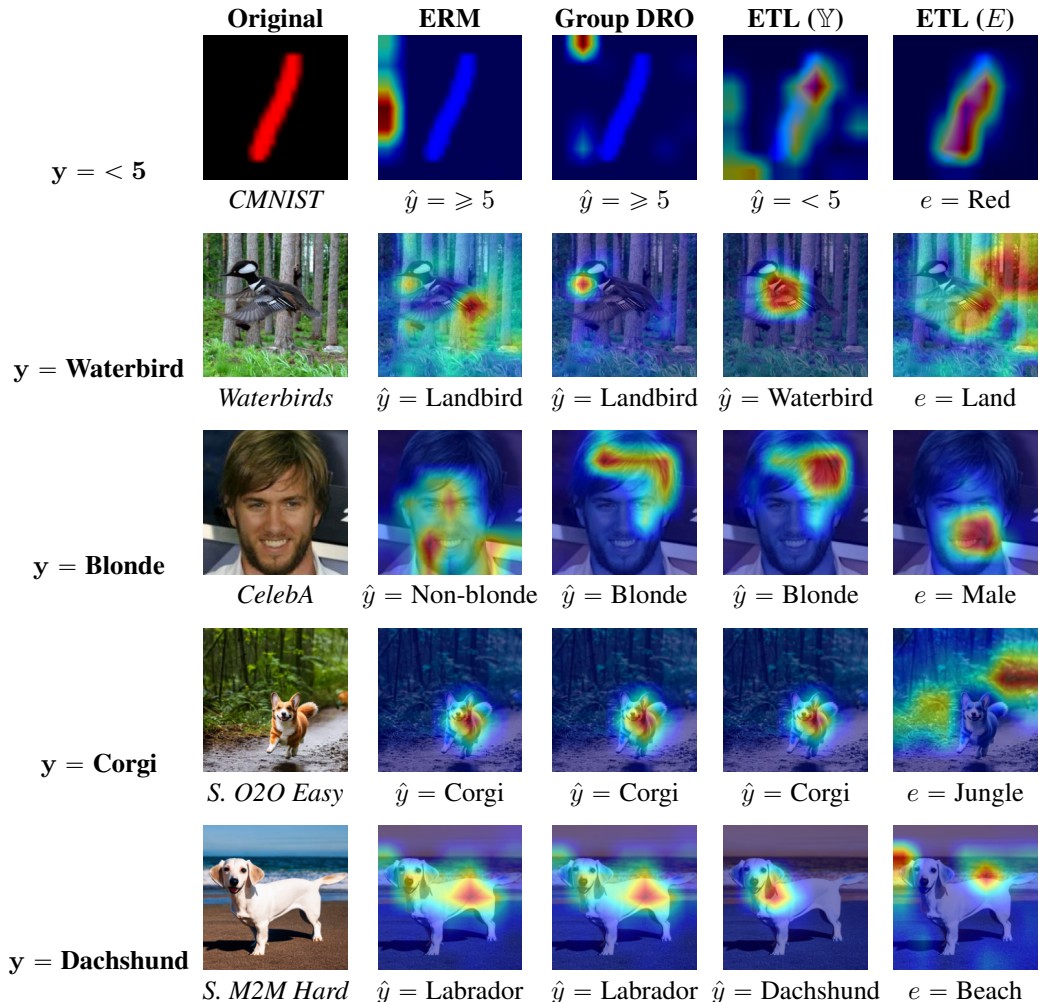

Figure 1: GradCAM plots with their corresponding true labels $y$, predicted labels $\hat{y}$, and true confounders $e$.

tion, while ETL provides a cleaner and more compact *check-like* label separation (extremely high $\text{MMD}_\mathbb{Y}$), presenting a more exact decision boundary than GroupDRO. Lastly, while ERM cleanly separates confounders hinting to confounder reliance in its predictions, ETL and GroupDRO shows interspersion of the confounders, pointing to non-reliance on confounders for their predictions, especially with ETL (extremely low $\text{MMD}_E$).

For Figure 3, domain invariance is evident in plots, when the environments are interspersed with each other and not forming defined clusters (low $\text{MMD}_{\text{Environment}}$). Interestingly, while ERM is already expected to not exhibit domain invariance as evidenced by its defined environmental clusters, ETL exhibits more interspersing of environments (lower $\text{MMD}_{\text{Environment}}$), as compared to a domain generalization SOTA, DANN, which is expected to perform better in promoting domain invariance.

## 6 CONCLUSION

In this paper, we presented Explaining to Learn (ETL), an interpretable explanation-based learning algorithm that removes spatial confounders from the primary classifier's latent representations during training. Its highly understandable visual explanations, coupled by its theoretical guarantees and performance gains over other SOTA methods, position it as a novel, and most importantly interpretable, high-performing algorithm in the field of distribution shifts. This opens up further studies

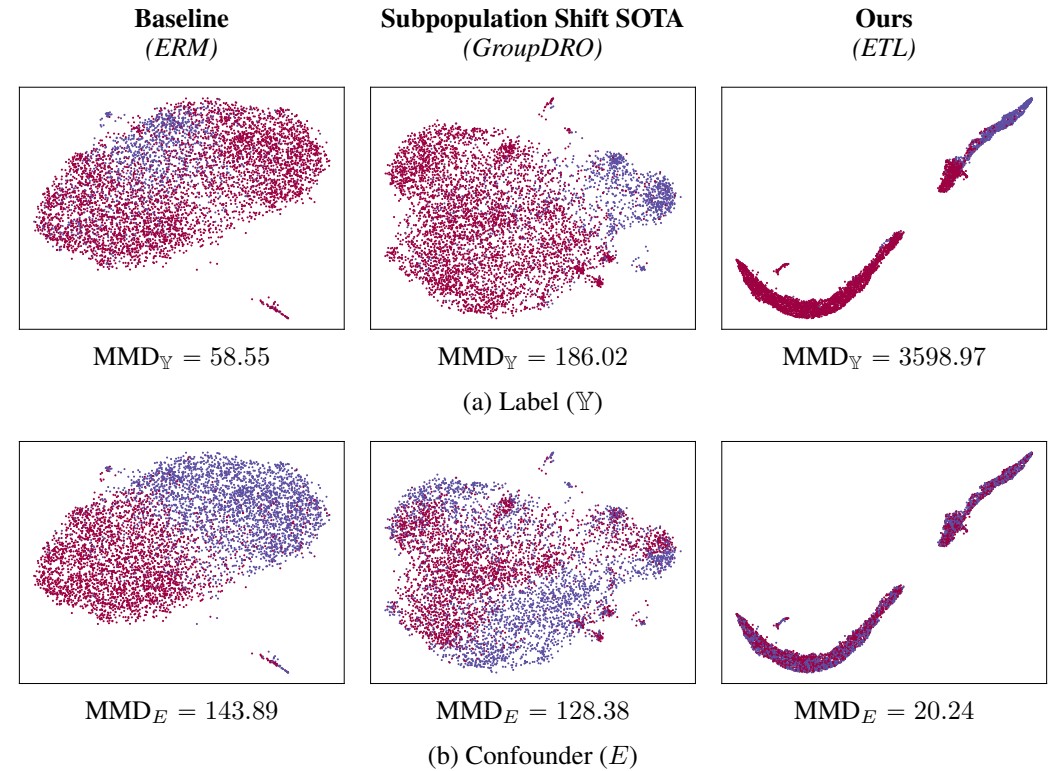

|  |  |  |
| :---: | :---: | :---: |
| **Baseline** | **Subpopulation Shift SOTA** | **Ours** |
| *(ERM)* | *(GroupDRO)* | *(ETL)* |

$\mathrm{MMD}_{\mathbb{Y}} = 58.55$     $\mathrm{MMD}_{\mathbb{Y}} = 186.02$     $\mathrm{MMD}_{\mathbb{Y}} = 3598.97$

(a) Label ($\mathbb{Y}$)

$\mathrm{MMD}_E = 143.89$     $\mathrm{MMD}_E = 128.38$     $\mathrm{MMD}_E = 20.24$

(b) Confounder ($E$)

Figure 2: UMAP Plots of different algorithms' latent representations on the *Waterbirds* dataset. (a) maps the color of each point by the label while (b) maps the color of each point by the confounder.

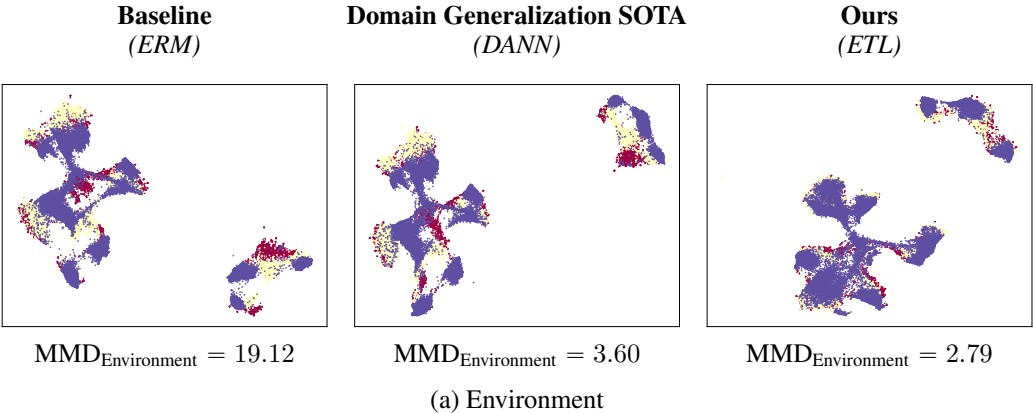

|  |  |  |
| :---: | :---: | :---: |
| **Baseline** | **Domain Generalization SOTA** | **Ours** |
| *(ERM)* | *(DANN)* | *(ETL)* |

$\mathrm{MMD}_{\text{Environment}} = 19.12$     $\mathrm{MMD}_{\text{Environment}} = 3.60$     $\mathrm{MMD}_{\text{Environment}} = 2.79$

(a) Environment

Figure 3: UMAP Plots of different algorithms' latent representations on the *Hard Many-to-Many Spawrious* dataset. (a) maps the color using the defined training and test environments, which are three in total.

in leveraging intersections in machine learning fields, in this case, the intersection of Distribution Shifts and Explainable AI (xAI).

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

## A APPENDIX

### A.1 METHODOLOGY DETAILS

#### A.1.1 DATASETS

**CMNIST.** CMNIST is a dataset based on the MNIST dataset, where the labels are divided into "Less than 5", and "Greater than or Equal to 5". (Arjovsky et al., 2020). Arjovsky et al. (2020) adds the spurious correlation, color, by coloring them red or green based on the label. This correlation is subverted at test time, where the spurious correlation is reversed (Arjovsky et al., 2020). Additionally, labels are flipped with a 0.25 probability, to add noise in the dataset (Arjovsky et al., 2020).

**Waterbirds.** Waterbirds is a dataset based on the CUB dataset, in which pictures of birds are spuriously correlated with the land or water background (Sagawa* et al., 2020). The challenge here is that birds which do not necessarily match with their background (e.g., Water Bird in Land) are minority groups, leading to the problem of subpopulation shift at test time, where they are balanced out again (Sagawa* et al., 2020).

**CelebA.** CelebA is based on the CelebA celebrity face dataset in which the labels are the hair color, and the confounders are the gender of the image subjects (Sagawa* et al., 2020). The spurious

correlation here is that most females are blondes, while most males are non-blondes (Sagawa* et al., 2020).

**Spawrious.** Spawrious is a newer dataset which tests models on unseen domains (mainly dogs and backgrounds), where it has two types and three difficulty (Easy, Medium, Hard) per type (Lynch et al., 2025). One type is "One-to-One" in which dogs are only associated with one background, and this background is changed at test time (Lynch et al., 2025). On the other hand, the "Many-to-Many" creates a many-to-many correlation in which dogs are associated with two backgrounds, and these two backgrounds are changed at test time (Lynch et al., 2025).

### A.1.2 ALGORITHMS

The authors referred to Gulrajani & Lopez-Paz (2021)'s implementation for these algorithms, running three tests for each algorithm with different random seed initializations to account for the initialization factor.

### A.1.3 MODELS

The authors used `PyTorch`'s implementation of `ResNet50` for all algorithms, using the default weights or `IMAGENET1K_V2`.

### A.1.4 TRAINING STRATEGY

Fixing Stochastic Gradient Descent (SGD) as the optimizer to promote comparability, hyperparameter tuning for each algorithm was done through a randomized grid search, using Gulrajani & Lopez-Paz (2021)'s hyperparameter distributions and using 20 randomized trials. Moreover, for the first run, the average of AA and WGA served as the early stopping criteria, while for the next runs, the full duration of the epochs were used. For all runs, model checkpointing is done under the condition that both AA and WGA improved.

### A.2 PROOFS

To prove the viability of ETL's $l_{sim}$, we now demonstrate its learning stability. Our reasoning proceeds from a set of mild assumptions to show that the loss function is *Lipschitz continuous* with respect to the model parameters, which is a key condition for stable gradient-based optimization.

**Assumption 1** *There exists an nth-indexed final-layer weight parameter $w_n^{\hat{t}_i}$ corresponding to a target logit $\hat{t}_i$ which is not equal to the nth-indexed final-layer weight parameter $w_n^{\hat{t}_j}$ corresponding to another target logit $\hat{t}_j$.*

$$\exists w_n^{\hat{t}_i} : w_n^{\hat{t}_i} \neq w_n^{\hat{t}_j}; i, j \in \{1, \ldots, |T|\}; i \neq j \tag{13}$$

Given Assumption 1 and a single input tensor **X**, by decomposing $\alpha_k^{\hat{t}}$ into $\frac{1}{P}$ and $w_k^{\hat{t}}$ using the Chain Rule, we can prove that the GradCAM maps of two logits $\hat{t}_i$ and $\hat{t}_j$ from the same classifier are also unequal.

**Lemma 1 (GradCAM Target-Specificity)** Given a single input tensor **X** and a model $m$ and using the chain rule on $\alpha_k^{\hat{t}_n}$, the GradCAM maps of any two logit from the same model and input are unequal.

Given that GradCAM is defined by:

$$L_{GradCAM}^{\hat{t}} = ReLU(\frac{1}{P} \sum_k \alpha_k^{\hat{t}} A^k) \tag{14}$$

Given that it is the same input, we can treat $A^k$ as fixed. Focusing more on the $\alpha_k^{\hat{t}}$, we know that it is defined as:

$$\alpha_k^{\hat{t}} = \frac{1}{P} \sum_{i,j} \frac{\partial \hat{t}}{\partial A_{i,j}^k} \tag{15}$$

Using the Chain Rule, we can compose it into two partial derivatives:

$$\frac{\partial \hat{t}}{\partial A_{i,j}^k} = \frac{\partial \hat{t}}{\partial Z_k} \frac{\partial Z_k}{\partial A_{i,j}^{\hat{t}}} \tag{16}$$

where, we can define $Z_k$ as:

$$Z_k = \frac{1}{k} \sum_k \sum_{i,j} A_{i,j}^k \tag{17}$$

Taking its partial derivative with respect $A_{i,j}^k$, we get:

$$Z_k = \frac{1}{k} \tag{18}$$

On the other hand, assuming a generalization based on Class Activation Mapping (CAM) (Selvaraju et al., 2019), we can define the logit $\hat{t}$ as:

$$\hat{t} = \sum_k w_k^{\hat{t}} Z_k + b_n \tag{19}$$

where $b_n$ is the bias term per neuron $n$.

We can take its partial derivative with respect to $Z_k$, such that:

$$\frac{\partial \hat{t}}{\partial Z_k} = k w_k^{\hat{t}} \tag{20}$$

Hence, we get:

$$\frac{\partial \hat{t}}{\partial A_{i,j}^k} = w_k^{\hat{t}} k \frac{1}{k} \tag{21}$$

$$\frac{\partial \hat{t}}{\partial A_{i,j}^k} = w_k^{\hat{t}} \tag{22}$$

Given that $P = k$, we can put it all together to state that:

$$L_{GradCAM}^{\hat{t}_i} \neq L_{GradCAM}^{\hat{t}_j} \tag{23}$$

$$ReLU(\frac{1}{P} \sum_k \alpha_k^{\hat{t}_i} A^k) \neq ReLU(\frac{1}{P} \sum_k \alpha_k^{\hat{t}_j} A^k) \tag{24}$$

Substituting the other definition to the neuron importance weights,

$$\alpha_k^{\hat{t}} = \frac{1}{P} \sum_{i,j} w_k^{\hat{t}} \tag{25}$$

Hence, given a fixed $A_k$ and Assumption 1,

$$ReLU(\frac{1}{P^2}\sum_k w_k^{\hat{t_i}} A^k) \neq ReLU(\frac{1}{P^2}\sum_k w_k^{\hat{t_j}} A^k) \tag{26}$$

This completes the proof.

Placing this in the label-confounder classifier pair setting, we assume the following:

**Assumption 3** *For a given input* **X***, the activation maps produced by the label model's feature extractor, $A_y^k$, and the confounder model's feature extractor, $A_e^k$, are not identical.*

$$A_y^k(\mathbf{X}) \neq A_e^k(\mathbf{X}) \tag{27}$$

Using Assumption 3, we can then extend Lemma 1 to the label-confounder classifier pair setting:

**Corollary 2 (GradCAM Target-Specificity Extension)** *Extending Lemma 1 for two models with Assumption 2, the GradCAM maps of any two logits each from the label $y$ and confounder $e$ model are unequal.*

*Given these conditions, we can directly substitute and state that:*

$$ReLU(\frac{1}{P^2}\sum_k w_k^{\hat{y}} A_y^k) \neq ReLU(\frac{1}{P^2}\sum_k w_k^{\hat{e}} A_e^k) \tag{28}$$

Given that we have established that GradCAM maps of any label logit $\hat{y}$ and confounder logit $e$ are unequal and are proportional to the logit final-layer weights, hence highlighting activation map importance, we can now define their similarity function.

**Definition 2 (GradCAM Similarity Function)** *A GradCAM Similarity Function, $S$, is a function which takes two inputs, GradCAM maps ($M_y$, $M_e$) from a label and classifier model. This function is $L$-Lipschitz continuous and differentiable for all $x$, or at least differentiable for a given interval and subdifferentiable at a given $x$.*

$$GradCAM\ Similarity = S(M_y, M_e) \tag{29}$$

Using this definition, we can then check the function's stability using the $L\text{-}Lipschitz\ continuity$ equation.

**Theorem 1. (Similarity Function Loss Stability)** Given Definition 2, we define the similarity function loss as $l_{sim}(\theta) = S(M_y, M_e)$ where $\theta$ is the model parameters, and using $L\text{-}Lipschitz\ continuity$ equation, we can prove that:

$$l_{sim}(\theta) - l_{sim}(\theta') \leqslant L\sqrt{C_y^2 + C_e^2} \cdot \|\theta - \theta'\|_2 \tag{30}$$

*Proof.* The change in the Grad-CAM map $M(\theta)$ is bounded by the change in model parameters $\theta$. There exists a constant $C_M$ such that:

$$\|M(\theta) - M(\theta')\|_F \leqslant C_M \cdot \|\theta - \theta'\|_2$$

Let $M(\theta) = ReLU\left(\sum_k \alpha_k(\theta)A^k(\theta)\right)$. Since ReLU is 1-Lipschitz:

$$\|M(\theta) - M(\theta')\|_F \leqslant \left\|\sum_k \alpha_k(\theta)A^k(\theta) - \sum_k \alpha_k(\theta')A^k(\theta')\right\|_F$$

$$\leqslant \left\|\sum_k (\alpha_k(\theta) - \alpha_k(\theta'))A^k(\theta)\right\|_F + \left\|\sum_k \alpha_k(\theta')(A^k(\theta) - A^k(\theta'))\right\|_F$$

$$\leqslant \sum_k |\alpha_k(\theta) - \alpha_k(\theta')| \cdot \|A^k(\theta)\|_F + \sum_k |\alpha_k(\theta')| \cdot \|A^k(\theta) - A^k(\theta')\|_F$$

We now prove the main theorem for the loss $\ell_{sim}(\theta) = S(M_y(\theta), M_e(\theta))$. By Definition 1, $S$ is $L_S$-Lipschitz continuous:

$$|\ell_{sim}(\theta) - \ell_{sim}(\theta')| = |S(M_y(\theta), M_e(\theta)) - S(M_y(\theta'), M_e(\theta'))|$$

$$\leqslant L_S \cdot \sqrt{\|M_y(\theta) - M_y(\theta')\|_F^2 + \|M_e(\theta) - M_e(\theta')\|_F^2}$$

Applying the result for both the label map ($M_y$, with constant $C_y$) and the confounder map ($M_e$, with constant $C_e$):

$$|\ell_{sim}(\theta) - \ell_{sim}(\theta')| \leqslant L_S \cdot \sqrt{(C_y \cdot \|\theta - \theta'\|_2)^2 + (C_e \cdot \|\theta - \theta'\|_2)^2}$$

$$= L_S \cdot \sqrt{C_y^2 + C_e^2} \cdot \|\theta - \theta'\|_2$$

By defining the final constant $L_{total} = L_S\sqrt{C_y^2 + C_e^2}$, we arrive at the final statement of the proposition:

$$|\ell_{sim}(\theta) - \ell_{sim}(\theta')| \leqslant L_{total} \cdot \|\theta - \theta'\|_2 \tag{31}$$

This completes the proof.

## A.3 SIMILARITY FUNCTION SELECTION TEST RESULTS

Table 3: Similarity Function Selection Test Results

| Algorithms | | Datasets | | | |
|---|---|---|---|---|---|
| | | CMNIST | | Waterbirds | |
| Name | Sampling | Train WGA | Val WGA | Train WGA | Val WGA |
| Cosine | Random | 42.51% | 8.52% | 15.04% | 18.05% |
| **Cosine** | **Uniform Group** | 72.38% | 64.13% | 91.55% | **85.71%** |
| IoU | Random | 5.55% | 0.84% | 43.48% | 44.64% |
| IoU | Uniform Group | 65.67% | 65.46% | 91.70% | 84.55% |
| JS Dist. | Random | 0.00% | 0.00% | 25.10% | 25.70% |
| **JS Dist.** | **Uniform Group** | 71.31% | **67.83%** | 88.20% | 82.71% |
| JS Div. | Random | 16.02% | 8.75% | 15.04% | 19.55% |
| JS Div. | Uniform Group | 71.19% | 66.22% | 90.68% | 79.70% |
| KL Div. | Random | 10.92% | 5.29% | 39.13% | 45.11% |
| KL Div. | Uniform Group | 68.44% | 66.30% | 89.41% | 82.71% |
| MAE | Random | 4.59% | 5.38% | 62.50% | 53.38% |
| **MAE** | **Uniform Group** | 72.20% | 66.43% | 89.93% | **85.71%** |
| MSE | Random | 4.90% | 2.09% | 16.07% | 18.05% |
| MSE | Uniform Group | 64.07% | 55.15% | 87.08% | 83.91% |
| NCC | Random | 0.39% | 0.30% | 17.88% | 18.05% |
| NCC | Uniform Group | 72.37% | 66.16% | 91.60% | 82.19% |
| RMSE | Random | 17.74% | 2.54% | 10.71% | 15.79% |
| RMSE | Uniform Group | 65.52% | 62.78% | 91.09% | 84.76% |
| SSIM | Random | 74.77% | 22.42% | 14.87% | 17.99% |
| **SSIM** | **Uniform Group** | 71.48% | **67.41%** | 89.97% | 81.55% |
| Soft Dice | Random | 15.34% | 6.43% | 46.43% | 40.60% |
| **Soft Dice** | **Uniform Group** | 69.79% | **66.52%** | 93.06% | **84.96%** |

## A.4 CELEBA RESULTS

Table 4: CelebA Experiment Results

| Algorithms | | Subpopulation Shift | |
| --- | --- | --- | --- |
| | | **CelebA** | |
| **Name** | **Sampling** | **AA** | **WGA** |
| *Baselines* | | | |
| ERM | Random | 95.56 ±0.0% | 39.63 ±2.6% |
| IRM | Uniform Env. | 91.17 ±0.0% | 82.04 ±1.8% |
| MMD | Uniform Env. | 92.19 ±0.3% | 81.30 ±1.4% |
| CORAL | Uniform Env. | 92.20 ±0.4% | 82.41 ±0.8% |
| DANN | Uniform Env. | 92.23 ±0.3% | 80.93 ±2.2% |
| CDANN | Uniform Env. | 90.16 ±0.2% | 80.56 ±2.2% |
| **GroupDRO** | **Uniform Group** | **91.94 ±0.3%** | **84.07 ±0.6%** |
| *Ours* | | | |
| ETL-MAE | Uniform Group | 90.79 ±0.2% | 82.59 ±2.0% |
| **ETL-Cosine** | **Uniform Group** | **91.19 ±0.3%** | **84.07 ±1.6%** |
| **ETL-Soft Dice** | **Uniform Group** | **91.70 ±0.3%** | **83.33 ±1.0%** |
| ETL-JS Dist. | Uniform Group | 90.05 ±0.5% | 84.07 ±1.7% |
| ETL-SSIM | Uniform Group | 91.39 ±0.2% | 80.93 ±2.7% |

