# OpenReview forum: "Explaining to Learn: Regularization Using Contrastive Visual Explanation Pairs For Distribution Shifts"
_ICLR.cc/2026/Conference — ICLR 2026 Conference Withdrawn Submission_

### Official Review · Reviewer_aTYW · 2025-10-27

**Soundness:** 2
**Presentation:** 1
**Contribution:** 2
**Rating:** 2
**Confidence:** 3

**Summary:**

This paper proposes Explaining to Learn (ETL), an explanation-based regularization method to improve robustness to distribution shifts by penalizing simialrity between heatmaps generated from GradCAM of a label classfier and a confound classifier. The authors claim that this approach explicitly removes spatial confounders and enhances interpretability. Experiments on CMNIST, Waterbirds and CelebA show superior performance over simple baseline such as GroupDRO.

**Strengths:**

1. The idea of incorporating GradCAM-based explanations directly into the training process to mitigate distribution shifts is novel and conceptually appealing.

2. Experimental results show consistent improvements across multiple benchmarks, including Waterbirds and CelebA, demonstrating the method’s effectiveness in diverse dataset.

3. The approach enhances interpretability by explicitly linking model decisions to visual explanations.

**Weaknesses:**

1. The training procedure is difficult to follow. The paper does not provide a concise, end-to-end description of the full training pipeline, and the reader have to reconstruct it from scattered equations and algorithm boxes.

2. The method depends on GradCAM, which is designed specifically for CNN-based architectures. This limits its generality and prevents application to modern architectures such as Vision Transformers (ViTs). No experiments are conducted with other attribution methods (e.g., Integrated Gradients or GradCAM++), which could help demonstrate whether the proposed idea generalizes beyond CNN settings.

3. Since GradCAM requires gradient computation with respect to intermediate feature maps, incorporating it in every training iteration can significantly increase both training time and memory cost. The paper does not quantify or analyze this additional overhead, making it unclear whether ETL remains efficient compared to simpler baselines such as GroupDRO.

4. The experimental evaluation omits several recent competitive methods yet proposed early, such as JTT proposed in 2021 [1] and DFR proposed in 2023 [2]. These methods achieve substantially higher worst-group accuracies on benchmarks like Waterbirds (e.g., DFR: 92.9%), where the performance of ETL is not competitve (ETL-MAE: 87.45%) while requiring more data annotation (confounder) and computation cost (GradCAM).

[1] Just Train Twice: Improving Group Robustness without Training Group Information, ICML 2021.

[2] Last Layer Re-Training is Sufficient for Robustness to Spurious Correlations, ICLR 2023.

**Questions:**

1. Would ETL still work with other attributions that can be generalize to other architectures (e.g. Integrated Gradient, Attention Rollout)?

2. Could the authors provide comparison with more advanced methods especially on spurious correlation?

3. What is the additional computation cost (FLOPs / clock-wise time) of ETL compared to vanilla baseline?

---

> ### Author Response · Authors · 2025-11-24
> **Response to Reviewer aTYW**
>
> **Response to Reviewer aTYW**
>
> Thank you very much for your thoughtful review and for recognizing both the value and novelty of the proposed ETL method. We address your questions and requests for clarification in the sections below.
>
> **I. Applicability of ETL to Other Attribution / Explainability Techniques**
>
> We agree that ETL, in principle, can be generalized to alternative attribution methods. Since ETL is based on comparing explanation maps, its framework is compatible with other techniques that produce spatially interpretable outputs.
>
> That said, our current choice of GradCAM is intentional and carefully grounded:
>
> > As noted in our response to Reviewer BCvE, Adebayo et al. (2018) showed that GradCAM is uniquely sensitive to model reparameterization and to input–label relationships, unlike other methods such as Integrated Gradients, DeepLIFT, or Guided Backprop, which often behave like unsupervised edge detectors [1].
>
> Because ETL uses explanation discrepancies as a training signal, sensitivity to parameter and concept changes is essential, making GradCAM particularly appropriate for ETL’s regularization mechanism.
>
> **II. Comparison with JTT and DFR**
>
> We appreciate the suggestion to compare ETL with JTT and DFR. Our decision not to include them as baselines stems from methodological considerations:
>
> **1. ETL as a regularization / loss-based method**
>
> - ETL belongs to the family of loss-based or regularization methods for handling distribution shifts. For this reason, our baseline comparisons focused on methods in the same category (e.g., GroupDRO, MMD, CORAL).
>
> **2. JTT and DFR as two-step fine-tuning strategies**
>
> JTT and DFR differ fundamentally in algorithmic structure.
>
> Both use a two-step pipeline:
>
> * Train a classifier with ERM.
>
> * Fine-tune on a curated dataset (JTT) or a group-balanced dataset (DFR) [2, 3].
>
> ETL, by contrast, uses a single-step regularization loss that directly penalizes similarity between label and confounder GradCAM maps.
>
> **3. Annotation requirements**
>
> While JTT requires fewer group annotations, its performance is typically below GroupDRO [2], which ETL outperforms on challenging benchmarks.
>
> DFR, although strong, implicitly also requires complete group annotations to construct a group-balanced dataset [3], the same information needed by GroupDRO and other annotation-dependent methods.
>
> Thus, although JTT and DFR are valuable techniques, they belong to a different category from ETL, and were therefore excluded from the baseline comparisons.
>
> We do, however, agree that combining ETL with the second-stage fine-tuning in JTT or DFR is an appealing avenue for future research, especially because these works currently rely only on ERM loss for both stages.
>
> **III. Computational Overhead**
>
> On the Spawrious Many-to-Many Difficult benchmark:
>
> * ETL requires 59 seconds per epoch,
>
> * while ERM requires 33 seconds per epoch.
>
> This increase is expected for gradient-dependent algorithms. Just as FISH (Shi et al., 2021) incurs additional computational cost due to gradient manipulation [4], ETL’s use of GradCAM-based regularization naturally increases training time.
>
> However, the trade-off is justified:
>
> * ETL consistently achieves substantial improvements in both average and worst-group accuracy.
>
> * These gains are especially pronounced on challenging spurious correlation benchmarks, where non-gradient-based methods such as MMD, GroupDRO, and CORAL typically struggle.
>
> Given these, thank you once again for your insightful comments and for recognizing the novelty of ETL. We will incorporate the requested clarifications, to strengthen the overall presentation of the method.
>
> **References:**
> * [1]  Adebayo, J., et. al. (2018). Sanity checks for saliency maps. In Proceedings of the 32nd International Conference on Neural Information Processing Systems.
> * [2] Just Train Twice: Improving Group Robustness without Training Group Information, ICML 2021.
> * [3] Last Layer Re-Training is Sufficient for Robustness to Spurious Correlations, ICLR 2023.
> * [4] Shi, Y., et al. (2022). Gradient Matching for Domain Generalization. In International Conference on Learning Representations (ICLR).

---

### Official Review · Reviewer_ghJX · 2025-10-30

**Soundness:** 2
**Presentation:** 2
**Contribution:** 2
**Rating:** 6
**Confidence:** 3

**Summary:**

This paper proposes Explaining to Learn, a novel training framework that leverages contrastive GradCAM explanations to remove spatial spurious correlations under distribution shifts. Instead of solely aligning representations or reweighting groups, ETL jointly trains a primary classifier and a confounder classifier, and penalizes similarity between their GradCAM activation maps. This encourages the model to rely on semantically meaningful regions while suppressing confounding visual cues.

Experiments on CMNIST, Waterbirds, CelebA, and the Spawrious benchmark show that ETL improves WGA and is particularly strong under complex many-to-many background shifts.

**Strengths:**

-Turning GradCAM into a training signal is fresh and intuitive. The paper bridges explainability and robustness research, opening a direction of explanation-guided learning.
- Explicitly tackles spatial spurious correlations: Directly regularizes where the model looks, rather than indirectly enforcing invariance.
- Significant margins on Hard Spawrious benchmark (+5% AA, +11% WGA)
- Clear motivation & narrative: Easy to follow the causal logic -- don’t look at confounder pixels.

**Weaknesses:**

- assumes environment/confounder annotations, which limits applicability in scenarios without such metadata.

- heavily on GradCAM accuracy. GradCAM is coarse and can be noisy or misleading in some architectures and tasks.

- computation overhead: Two models + GradCAM computation increases training cost.

- focuses only on spatial confounders: may not generalize to style, texture, frequency, temporal, or high-level semantic spurious cues.

**Questions:**

see Weakness

---

> ### Author Response · Authors · 2025-11-24
> **Response to Reviewer ghJX**
>
> **Response to Reviewer ghJX**
>
> Thank you for your thoughtful and constructive review of our work. We appreciate your recognition of the paper’s strengths and your comments on aspects that would benefit from additional clarification. We respond to each of your points below.
>
> **I. Assumption on the Existence of Environment / Confounder Annotations**
>
> We acknowledge that ETL requires confounder presence annotations. However, we would like to clarify that this requirement is not unique to our method and is in fact shared with widely used approaches for addressing shortcut learning.
>
> GroupDRO and LISA, two prominent algorithms in this space, explicitly rely on environment or group annotations to mitigate biases [1, 2]. Despite this annotation overhead, these methods are widely used due to their strong performance, something that ETL similarly delivers, especially on difficult benchmarks such as Spawrious Many-to-Many Difficult.
>
> Finally, as discussed in Dammu and Shah’s semi-supervised approach, even when detection methods attempt to minimize supervision, the final step still requires human verification to confirm that detected patterns are indeed spurious [3]. This reinforces the broader consensus that human-in-the-loop annotation remains necessary to ensure correctness and reliability in debiasing pipelines.
>
> Thus, while ETL requires confounder tags, this requirement is consistent with existing methods, and the performance gains of ETL outweigh the annotation overhead.
>
> **II. Reliance on GradCAM Accuracy**
>
> We agree with the reviewer that ETL’s effectiveness depends partly on the faithfulness of GradCAM. We offer three key clarifications:
>
> **1. This reliance is inherent to all gradient-dependent methods.**
> - For example, FISH (Shi et al., 2022) similarly depends on the stability and accuracy of gradients [4]. This is not a limitation unique to ETL.
>
> **2. Empirically, ETL remains robust.**
> - Across multiple random seeds, ETL consistently demonstrates strong performance. This indicates that even if GradCAM were imperfect in theory, the method is stable in practice.
>
> **3. Our choice of GradCAM is evidence-based.**
> - As discussed in our response to Reviewer BCvE, we rely on the analysis of Adebayo et al. (2018) [5], who show that GradCAM is:
>
>    * sensitive to model reparameterization, and
>
>    * sensitive to input–label relationships,
>
> which are essential properties for methods that use explanations as training signals.
>
> In contrast, alternative explanation methods (e.g., Integrated Gradients, DeepLIFT, $\epsilon$-LRP, Guided Backprop) lack these sensitivities and behave more like unsupervised edge detectors, making them unsuitable for ETL.
>
> Taken together, these points support our decision to use GradCAM.
>
> **III. Computational Overhead of ETL**
>
> We acknowledge, as discussed in our response to Reviewer BCvE, that ETL incurs additional computation time due to its reliance on gradient-based explanation maps. However, this overhead is expected and justified:
>
> * Gradient-dependent algorithms across the distribution shift literature, including FISH [4], are inherently more computationally demanding than non-gradient–based baselines.
>
> * Thus, while ETL is more computationally intensive than some alternatives, it provides substantial improvements in robustness and generalization, making the trade-off meaningful and worthwhile.
>
> **IV. ETL’s Focus on Spatial Confounders**
>
> We appreciate the reviewer’s point and would like to clarify our intent:
>
> > In alignment with the “No Free Lunch” principle, we see this specialization not as a limitation but as an important characteristic. It helps practitioners and researchers determine when ETL is the appropriate tool, depending on the nature of the confounders in their particular domain.
>
> By clearly positioning ETL as a method tailored for spatial confounding, we aim to support informed method selection in real-world applications.
>
> We thank the reviewer again for their constructive comments. We will incorporate these clarifications in the revised manuscript to better communicate the methodological assumptions, design choices, and intended scope of ETL.
>
> **References:**
> * [1] Sagawa, S., et. al. (2019). Distributionally robust neural networks for group shifts: On the importance of regularization for worst-case generalization.
> * [2] Yao, H., et. al. (2022, June). Improving out-of-distribution robustness via selective augmentation. In International Conference on Machine Learning.
> * [3] Dammu, P. P. S. & Shah, C. (2023). Detecting spurious correlations via robust visual concepts in real and AI-generated image classification. In XAI in Action: Past, Present, and Future Applications.
> * [4] Shi, Y., et al. (2022). Gradient Matching for Domain Generalization. In International Conference on Learning Representations (ICLR).
> * [5] Adebayo, J., et. al. (2018). Sanity checks for saliency maps. In Proceedings of the 32nd International Conference on Neural Information Processing Systems.

---

### Official Review · Reviewer_Ui92 · 2025-10-31

**Soundness:** 1
**Presentation:** 1
**Contribution:** 2
**Rating:** 2
**Confidence:** 5

**Summary:**

The authors propose a twin network, trained in parallel for mitigating bias and spurious correlation effects. One network learns the spurious attribute while the other one learns the real task label. During training, they compute the grad-CAM heatmaps for both networks and enforce them to be dissimilar. This simple, yet effective strategy seems to be capable of removing bias on different datasets. The proposed method outperforms all Domain Generalization baselines that perform on environments while it is also better than GroupDRO in most settings.

**Strengths:**

The proposed method is intuitive, simple, yet powerful. The idea is good enough to be worth a publication but the experimental setup and baseline choice are fragile. The choice of relevant and interesting bias mitigation datasets, including using multiple of them, is a strong point. Providing the code to let the reviewers and readers understand their implementation details is highly appreciated.

**Weaknesses:**

While the idea is very lean and useful, the execution and contextualization of this work have room for improvement. This idea has potential, but it is not fully explored in this submission.

- Contextualization: the idea is mainly framed in the field of distribution shifts, especially in the context of domain generalization. This, however, is not only limited but also misleading or simply wrong. There is tons of work in the field of bias mitigation and shortcut removal that are not considered at all. The only really relevant baseline method is GroupDRO that operates on a group level, thus being actually capable of removing biases/shortcuts effectively.

- Baselines: Related to the previous point, we also see that the choice of baselines in their experiments is heavily biased and mismatched. Instead of comparing against true bias mitigation and shortcut removal methods, the authors compare mainly against domain generalization methods. Kaur et al. [1] have shown that explicit methods that have access to groups outperform environment (domain) based methods. They also directly use a MMD-based approach that operates on groups (but priorly, other works already proposed MMD on groups, such as [2], [3]). Instead of using DANN, the authors should have further used a method that debiases the confounding attribute instead of the domain, such as [4,5, ...]. One can continue this list with a myriad of baselines that would have been more appropriate than the environment-based methods. Thus, these experiments are not convincing, as it is expected that group-based methods outperform these significantly [1].

- Unnecessary theoretical analysis: Subsection 3.3 feels bloated. The main message is simple and clear, and making the assumptions explicit is appreciated. Apart from that, there is a limited benefit in this subsection, since there are not concrete bounds for the Lipschitz analysis presented in the manuscript: it is well-known that chaining classical deep learning activations and matrix operations together with an explicitly defined Lipschitz function (Def. 1) yields another (locally) Lipschitz function. Without giving bounds, all this analysis is unnecessary as it was clear from the beginning that the regularization penalty will be smooth enough (as it already is differentiable, operates with well-known functions, etc.).

- Confusing notation and intransparency: it is unclear how the environments and the confounding variables relate. Without the code, this reviewer would not have been able to understand how the authors defined the different training and testing envs for the domain generalization methods. This adds a layer of intransparency and made the paper hard to follow.


[1] Kaur, Jivat Neet, Emre Kiciman, and Amit Sharma. "Modeling the Data-Generating Process is Necessary for Out-of-Distribution Generalization." The Eleventh International Conference on Learning Representations.
[2] Makar, Maggie, et al. "Causally motivated shortcut removal using auxiliary labels." International Conference on Artificial Intelligence and Statistics. PMLR, 2022.
[3] Veitch, Victor, et al. "Counterfactual invariance to spurious correlations in text classification." Advances in neural information processing systems 34 (2021): 16196-16208.
[4] Mitigating Unwanted Biases with Adversarial Learning
[5] Adeli, Ehsan, et al. "Representation learning with statistical independence to mitigate bias." Proceedings of the IEEE/CVF winter conference on applications of computer vision. 2021.

**Questions:**

Could the authors add highly relevant bias mitigation and shortcut removal methods introduced in the weaknesses section in an extensive comparison? This would make the entire work more credible.

---

> ### Author Response · Authors · 2025-11-24
> **Response to Reviewer BCvE**
>
> **Response to Reviewer Ui92**
>
> Thank you for your constructive feedback and for recognizing both the simplicity and empirical effectiveness of ETL. Your comments on improving the exposition of the experimental setup are well taken, and we outline our planned revisions and specific responses below.
>
> **I. On Contextualization and Baselines**
>
> We appreciate your concerns regarding the fairness of our baseline comparisons and the role of environment formulation.
>
> **1. Alignment with Kaur et al. (2024)**
>
> We agree with Kaur et al.’s observation that the performance of domain generalization algorithms is closely linked to the underlying data generation process, specifically, the degree to which environments differ in their label-confounder associations [1]. Algorithms struggle when environments are too similar, and perform better when these distributions are more distinct. This is very evident when Kaur, et. al. viewed the performance of the domain generalization algorithms in the small NORB dataset and found that these algorithms performed comparably with their algorithm in the environments of the NORB dataset, as they acted similarly to groups [1].
>
> **2. Ensuring fairness in our experimental design**
>
> To avoid inadvertently disadvantaging domain generalization (DG) methods, we carefully designed environments in each dataset to reflect meaningful differences in confounder-label distributions:
>
> **Waterbirds:**
> - Each environment is biased toward a specific scenery type (land vs. water), encouraging invariance across backgrounds.
>
> **CelebA:**
> - Environments are biased toward specific genders, paralleling standard subpopulation shift setups.
>
> **CMNIST:**
> - Each environment is biased toward a specific color (e.g., red or green), encouraging invariance across colors.
>
> Empirically, in our experiments, DG algorithms performed at levels comparable to, or in some cases outperforming, GroupDRO, which is a strong baseline designed for subpopulation shifts/bias mitigation and optimizes for the worst-performing group. This observation supports the claim that our environment formulation did not artificially suppress DG performance.
>
> **3. Applicability of DG methods to Spawrious benchmarks**
>
> The Spawrious dataset is a dataset in the domain generalization setting which contains domain shifts (e.g., chihuahuas seen only in beaches during training, but in snow during testing) and spurious correlation elements (e.g., chihuahuas being inevitably associated to beaches). This motivates the inclusion of both:
>
> * domain generalization algorithms, which aim to handle unseen domains, and
>
> * subpopulation shift algorithms, which aim to reduce reliance on spurious features.
>
> Thus, evaluating both classes of algorithms in this case is methodologically sound.
>
> **4. Competitiveness of ETL**
>
> Given the above, we maintain that:
>
> * the baselines used are appropriate and fairly evaluated,
>
> * the experimental design does not bias against DG algorithms, and
>
> * the results clearly show that ETL provides competitive and robust performance, particularly in the difficult Spawrious benchmarks, where it outperforms GroupDRO, a method specifically designed to excel under worst-group conditions.
>
> **5. Regarding additional baselines**
>
> We acknowledge that including the additional baselines you suggested would strengthen the paper. Nevertheless:
>
> * their absence does not invalidate the key contribution, and
>
> * the current baseline set is representative of both subpopulation and domain generalization methods, where proper experimental setup and data generation methods were in place to provide fair comparisons when using these methods.
>
> We will include a discussion of these points, especially on environmental formulation, in the revised manuscript to improve contextual clarity.
>
> Thank you once again for your valuable feedback. Your comments directly contribute to improving the clarity and rigor of the manuscript, and we believe the revisions will significantly strengthen the paper, allowing us to illustrate clearly the value of bridging explainability techniques with distribution shift methodologies.
>
> **References**
> * [1] Kaur, J. N., Kiciman, E., & Sharma, Amit. Modeling the Data-Generating Process is Necessary for Out-of-Distribution Generalization. In The Eleventh International Conference on Learning Representations.

---

### Official Review · Reviewer_BCvE · 2025-11-01

**Soundness:** 3
**Presentation:** 2
**Contribution:** 2
**Rating:** 4
**Confidence:** 4

**Summary:**

This paper proposes a method that uses GradCAM similarity penalties between a label classifier and confounder classifier during training. The goal is to remove spatial confounders from latent representations. The method penalizes similarity between GradCAM activation maps from both classifiers. Experiments show improvements on CMNIST, Waterbirds, and Spawrious benchmarks compared to baselines like GroupDRO.

**Strengths:**

1. Shows improvements on multiple benchmarks. Particularly strong on Spawrious Many-to-Many Hard benchmark (66.31% WGA vs 54.40% for GroupDRO).

2. Tests multiple similarity functions. Includes GradCAM visualizations and UMAP plots showing confounder separation.

3.  Paper is well-written. GradCAM visualizations effectively show the method addresses spatial confounders.

**Weaknesses:**

1. Using explanation-based methods for removing confounders is not new. The paper's own related work cites Hagos et al. (2022) and Dammu & Shah (2023) who use explainability for spurious correlations. The core idea of matching/penalizing explanation maps has been explored before.

2. Only works for spatial confounders in images. Cannot handle non-spatial spurious correlations.

3. Lipschitz continuity proof is trivial. Just shows loss is stable, doesn't explain why penalizing GradCAM similarity removes confounders or improves OOD generalization.

4. Assumption 2 (different activation maps) is critical but never validated empirically. May not hold in practice.

5. Trains two full networks plus computes GradCAM at every step. Paper doesn't report actual training time.

**Questions:**

1. Hagos et al. (2022) already used explanation-based learning for spurious correlations. How is ETL fundamentally different?

2. Why GradCAM specifically?: Other explanation methods exist. Why is GradCAM the right choice? Did you compare to other explanation methods?

3. What's the actual wall-clock training time compared to baselines?

---

> ### Author Response · Authors · 2025-11-24
> **Response to Reviewer BCvE**
>
> **Response to Reviewer BCvE**
>
> Thank you very much for your thoughtful review of our paper. We appreciate your recognition of the paper’s strengths, as well as your insightful comments regarding its positioning within explanation-based learning and distribution shift research. Below, we provide our responses to each of your points.
>
> **I. Distinguishing ETL from the Works of Hagos et al. (2022) and Dammu & Shah (2023)**
>
> To clarify how ETL differs from prior related work, we summarize each approach and then articulate the key distinctions.
>
> **Hagos et al. (2022)**
>
> - Hagos et al. employed a two-step procedure using a single classifier, initially trained on a spuriously correlated dataset and subsequently fine-tuned by comparing GradCAM maps against ground-truth mask annotations [1]. Their method requires expensive segmentation masks, and their focus lies primarily in reducing reliance on spurious features [1], without explicit attention to the downstream impact on overall classifier performance.
>
> **Dammu & Shah (2023)**
>
> Dammu and Shah (2023) proposed a technique aimed at detecting, rather than mitigating, spurious correlations [2]. Their process involves:
>
> * Training a classifier on the spuriously correlated dataset.
>
> * Extracting gradients, activations, and misclassification labels for each data point.
>
> * Training local surrogate models whose Shapley-value-weighted gradients are used to construct *RF-GradCAM* maps.
>
> * Comparing these maps with the classifier’s GradCAM maps to identify data points likely affected by spurious correlations, primarily for human verification, not automated debiasing.
>
> **How ETL Differs**
>
> Our work makes three main contributions relative to these two approaches:
>
> **1. Focus on correcting spurious correlations while improving classifier performance.**
> - Unlike Hagos et al.'s work, ETL does not require costly pixel-level masks; it relies only on simple confounder tagging [1]. Moreover, ETL explicitly demonstrates both reduction in spurious reliance and performance improvements, which Hagos et al. do not evaluate.
>
> **2. A simpler and more direct debiasing mechanism.**
> - While Dammu & Shah focus on detection and require surrogate modeling plus Shapley-value estimation [2], ETL directly aligns the explanations of a label classifier and a confounder classifier using GradCAM map comparison, avoiding the additional overhead of surrogate models.
>
> **II. Rationale for Using GradCAM over Other Explainability Techniques**
>
> We appreciate the reviewer’s interest in the choice of GradCAM. Our selection is grounded in findings from Adebayo et al. (2018) [3], who demonstrate that:
>
> * GradCAM is sensitive to model reparameterization and changes in the data-generation process [3].
>
> * Many backpropagation-based methods (Guided Backprop, Guided GradCAM) are invariant to higher-layer parameters, making them insufficient for ETL, where adjustments at higher layers must be reflected in the explanation maps [3].
>
> * Input-gradient–based techniques (e.g., Integrated Gradients, DeepLIFT, $\epsilon$-LRP) are largely sensitive only to the input edges, behaving similarly to unsupervised edge detectors, and do not adequately reflect input–label relationships, which are central to ETL’s training objective [3].
>
> The sensitivity of GradCAM to changes in both network parameters and data relationships is essential for ETL: it ensures that the explanation alignment loss meaningfully captures changes induced by training.
>
> **III. Comparison of ETL Training Time with Baseline**
>
> When trained on an A100 GPU using Spawrious Many-to-Many Difficult benchmark:
>
> * ETL trains at 59 seconds per epoch, compared to
>
> * ERM at 33 seconds per epoch.
>
> This increase in training time is expected. As with other gradient-based algorithms in the distribution-shift literature, such as FISH (Shi et al., 2021) [4], which also involves two interacting models, ETL incurs additional computational overhead due to its use of gradient-based explanation signals.
>
> However, we want to emphasize that performance over efficiency is the primary motivation of the paper. Our results show that this added cost yields significant gains, particularly on challenging benchmarks.
>
> We hope these clarifications help illustrate the novelty and rationale behind ETL.
>
> **References:**
> * [1] Hagos, M. T., et. al. (2022) Identifying spurious correlations and correcting them with an explanation-based learning. In NeurIPS 2022 workshop on Human-in-the-Loop Learning (HILL).
> * [2] Dammu, P. P. S. & Shah, C. (2023). Detecting spurious correlations via robust visual concepts in real and AI-generated image classification. In XAI in Action: Past, Present, and Future Applications.
> * [3] Adebayo, J., et. al. (2018). Sanity checks for saliency maps. In Proceedings of the 32nd International Conference on Neural Information Processing Systems.
> * [4] Shi, Y., et al. (2022). Gradient Matching for Domain Generalization. In International Conference on Learning Representations (ICLR).

---

### Note · Authors · 2026-01-24

I have read and agree with the venue's withdrawal policy on behalf of myself and my co-authors.